# PETrans: De Novo Drug Design with Protein-Specific Encoding Based on Transfer Learning

**DOI:** 10.3390/ijms24021146

**Published:** 2023-01-06

**Authors:** Xun Wang, Changnan Gao, Peifu Han, Xue Li, Wenqi Chen, Alfonso Rodríguez Patón, Shuang Wang, Pan Zheng

**Affiliations:** 1College of Computer Science and Technology, China University of Petroleum (East China), Qingdao 266580, China; 2Department of Artificial Intelligence, Faculty of Computer Science, Polytechnical University of Madrid, Campus de Montegancedo, Boadilla del Monte, Madrid 28660, Spain; 3Department of Accounting and Information Systems, University of Canterbury, Christchurch 8041, New Zealand

**Keywords:** drug discovery, de novo drug design, deep learning, transfer learning, molecule generation

## Abstract

Recent years have seen tremendous success in the design of novel drug molecules through deep generative models. Nevertheless, existing methods only generate drug-like molecules, which require additional structural optimization to be developed into actual drugs. In this study, a deep learning method for generating target-specific ligands was proposed. This method is useful when the dataset for target-specific ligands is limited. Deep learning methods can extract and learn features (representations) in a data-driven way with little or no human participation. Generative pretraining (GPT) was used to extract the contextual features of the molecule. Three different protein-encoding methods were used to extract the physicochemical properties and amino acid information of the target protein. Protein-encoding and molecular sequence information are combined to guide molecule generation. Transfer learning was used to fine-tune the pretrained model to generate molecules with better binding ability to the target protein. The model was validated using three different targets. The docking results show that our model is capable of generating new molecules with higher docking scores for the target proteins.

## 1. Introduction

The discovery of protein-targeted drugs is a very laborious, time-consuming and expensive process. Traditional methods such as high-throughput screening are inefficient because the number of resources required is not balanced by the small number of hit compounds. Conventionally, the identification of promising lead structures is achieved by experimental high-throughput screening (HTS), but this is time-consuming and expensive [1,2,3]. A typical drug discovery cycle takes approximately 14 years [4] and costs approximately 800 million dollars [5]. Another approach generates a large number of molecules and then uses virtual screening (VS) to find new ligands with desired properties. The ligands are typically sorted by their docking score to the receptor. However, this approach is not stable and does not generate target-specific molecules. Therefore, it is important to overcome the limitations of conventional drug discovery methods with efficient, cost-effective, and broad-spectrum computational alternatives [6]. With the development of computer technology, drugs were modeled and refined using machine learning based techniques. In recent years, artificial intelligence (AI) combined with computational chemistry offers a significantly more efficient alternative [7,8,9,10]. For example, the use of machine learning-based techniques to predict the inhibitory activity of high-value compounds has accelerated the identification of new drug candidates for COVID-19 [11]. At the same time, deep learning has been successfully applied to generating small molecules [12].

In 2006, Hinton et al. [13] officially proposed the concept of deep learning, which has already been successfully applied in computer vision, natural language processing, and some other fields [14,15,16,17,18]. In recent years, deep learning has been successfully applied to molecule generation. There are increasing efforts to develop deep learning algorithms that can automatically generate chemically valid molecular structures [19]. Similar to natural language processing and social networks, molecules are represented as texts and graphs [20,21,22]. Therefore, models for de novo molecular design are naturally applicable to drug discovery. For example, Zhavoronkov et al. [23] proposed the GENTRL algorithm to design potential molecules of DDR1 kinase inhibitors in just 21 days, significantly reducing the development time and cost of new drugs. Meanwhile, neural network-based molecule generators using a Variational Autoencoder (VAE) and a recurrent neural network (RNN) have been proposed for de novo molecule generation. Character VAE uses kernel density estimation to learn the relevant features of molecules. Then, a gradient-based search is used to efficiently guide the search by learning continuous latent spaces on dimensions, that optimize the specific properties of molecules. The molecule was decoded along a random start point (objective function value of 18.06%) to the end point (objective function value of 98.23%) using Gaussian interpolation [24]. Segler et al. attempted to pretrain the RNN neural network model on a generic dataset and then apply the pretrained model to a specific dataset through transfer learning to improve the prediction performance on a small dataset. Using this strategy, the model reproduced 14% of 6051 test molecules against Staphylococcus aureus and 28% of 1240 test molecules against Plasmodium falciparum (malaria) [25].

Novel drugs with potential therapeutic interactions are essential in the molecule generation process. One of the strategies focused on the simplified molecular input line entry system (SMILES) [26] that uses textual features to generate molecules. It is possible to generate input vectors from molecular structures, but reversing these vectors is extremely difficult, particularly because a single fingerprint represents multiple possible chemical structures. Using deep generative models with SMILES strings as molecular representations can help overcome this limitation. By converting molecular structures into texts, SMILES can be easily processed by computers, which is convenient for chemists and easy to train deep generative models [27,28]. Grechishnikova used the transformer architecture to consider target-specific de novo drug design as a translation problem between the amino acid “language” and the SMILES representation of the molecule [29]. However, approximately 30% of the generated molecules were found in the training dataset. Recently, Xu et al. [30] attempted to combine VAE models with docking score to generate molecules. However, this approach may result in a longer time to find the optimal molecule, thus prolonging the overall generation process. Some methods incorporated structural features to generate molecules. However, the limited quantity of known protein structures restricts the application of structure-based prediction methods. Since some protein structures are unknown, structural features rely on third-party computational tools that produce noise information that impacts predictions. Cell growth, survival and differentiation are regulated by receptor tyrosine kinases (RTKs) [31]. Among the known RTKs, the epidermal growth factor receptor has been widely studied. Overexpression of EGFR has been associated with aggressive disease and poor prognosis in a number of tumor types (e.g., breast, lung, ovarian, prostate, and squamous carcinoma of head and neck) [32,33]. However, there are few drugs targeting EGFR. Existing methods face the problem of data scarcity in the development of drugs against EGFR. As a critical sphingolipid metabolite, sphingosine-1-phosphate (S1P) plays an essential role in immune and vascular systems [34]. 5-HT receptors have high levels of basal activity and are subject to regulation by lipids, but the structural basis for the lipid regulation and basal activation of these receptors and the pan-agonism of 5-HT remains unclear [35].

For drug discovery and development, predictions of small molecules binding to proteins are especially useful for screening virtual libraries of drug-like molecules. Predicting bound conformations and binding affinity is the purpose of molecular docking [36]. To avoid potentially harmful side effects and reduce costs, the pharmaceutical industry is focusing on developing highly selective drugs using molecular modeling techniques [37,38]. Therefore, the process of molecular docking is crucial.

In this work, we pretrained a model based on GPT architecture on the MOSES dataset, and SMILES was used to represent molecules. Protein encodings are used to represent protein information. Transfer learning was applied to our model. We later applied the method on three targets (EGFR, S1PR1 and HTR1A) to investigate the capability of the PETrans to generate molecules with better binding ability to the target proteins. Our work aims to use deep learning to achieve de novo drug design for a specific target protein. In addition to drug repurposing, we provide new ideas for drug discovery.

In summary, our main contributions are: (i) adding protein-encoding information, such as amino acid composition information, amino acid order information, and physicochemical information, during the pretraining of the GPT model; (ii) generating molecules with higher docking scores with similar structures to the known drugs by transfer learning; (iii) generating novel molecules with high docking scores for three targets (corresponding to EGFR, S1PR1 and HTR1A), and the drug potentials of the generated molecules are also better than those of the known active compounds.

## 2. Results

To evaluate the generation performance of PETrans, we compared PETrans with SBMolGen, a deep learning-based molecular generator. SBMolGen integrates a recurrent neural network and Monte Carlo tree search [39]. We evaluated the general properties, docking scores and similarity scores of the generated molecules. The similarity scores measure the similarity between the generated molecules and the active compounds based on the same scaffold. We also conducted experiments involving the incorporation of protein secondary structure information and control of docking scores. To demonstrate the binding of the generated molecules to the pockets of the target proteins, we constructed binding models for ten molecules. In the following sections, we present our analysis of the experimental results in detail.

### 2.1. Evaluation Properties of the Generated Molecules

QuickVina-W Score estimates the binding affinity between the generated molecules and the protein pocket. The molecular docking mechanism between the generated molecules and three targets was investigated by AutoDock Vina program. We used QuickVina-W [40] to calculate the docking score for the protein and the generated molecules. QuickVina-W is a new docking tool that is suitable for large search spaces, especially for blind docking. In addition to the powerful scoring capabilities of AutoDock Vina, QVina-W incorporates the accelerated search capabilities of QVina 2 to provide detailed searching for large search spaces. QuickVina-W is faster than Quick Vina 2, and better than AutoDock Vina. Researchers can screen ligand libraries virtually, quickly, and accurately without having to define a target pocket in advance [40].

Quantitative Estimate of Drug-Likeness (QED) is a quantitative estimate of drug similarity that measures the probability that a molecule is a potential drug candidate.

Synthetic Accessibility Score (SA score) is the synthetic accessibility score that represents the difficulty of drug synthesis. It was normalized to a score between 0 and 1, where a higher score means that the molecule is easier to synthesize.

LogP stands for octanol–water partition coefficient and generally logP values between −0.4 and 5.6 are a good candidate for the drug.

Molecular Weight (MW) is the sum of the atomic weights of molecules. Drug-like molecules have molecular weights between 200 and 700 Da.

### 2.2. QuickVina-W Scores of the Generated Molecules

We randomly selected a set of 1350 molecules, equal in size to the transfer learning dataset. None of the 1350 generated molecules were found in the transfer learning dataset. They also did not appear in the MOSES dataset. The ten molecules with the lowest docking energy are shown in Figure 1. The binding models of the three molecules are shown in Figure 2. The distributions of the docking scores of the generated molecules for EGFR, S1PR1 and HTR1A are shown in Figure A1. Examples of molecules generated for other targets are shown in Figure A2 and Figure A3.

We selected three generated molecules to demonstrate their binding models. The binding models of the three molecules are shown in Figure 2. The left side of Figure 2 is the structure of the generated molecules, and the docked complex structure was created by PyMOL [41] (middle of Figure 2) and CB-Dock2 [42,43] (right side of Figure 2).

According to the docking results, the x, y, z centers are 23, 38, and 87, and the active residues of the receptor were revealed as follows: LEU718, GLY719, SER720, GLY721, VAL726, ALA743, ILE744, LYS745, MET766, CYS775, ARG776, LEU777, LEU788, THR790, PRO794, PHE795, GLY796, CYS797, ASP800, TYR801, GLU804, ARG841, ASN842, LEU844, THR854, ASP855, PHE856, LEU858, ASN996, TYR998, LEU1001, and MET1002(docking score: −9.8 kcal/mol); LEU718, GLY719, GLY721, VAL726, ALA743, ILE744, LYS745, MET766, CYS775, ARG776, LEU777, LEU788, THR790, PHE795, GLY796, CYS797, ASP800, TYR801, GLU804, ARG841, ASN842, LEU844, THR854, ASP855, PHE856, LEU858, ASN996, TYR998, LEU1001, and MET1002(docking score: −9.6 kcal/mol); LEU718, GLY719, SER720, GLY721, ALA722, VAL726, ALA743, ILE744, LYS745, MET766, CYS775, ARG776, LEU777, LEU788, ILE789, THR790, GLN791, LEU792, MET793, PHE795, GLY796, CYS797, ARG841, ASN842, LEU844, THR854, ASP855, PHE856, and LEU858 (docking score: −9.5 kcal/mol).

### 2.3. The Comparison of Properties

The mean values of the docking score, QED, SA score and LogP are shown in Table 1. We selected three molecules from each of the three different sets to demonstrate their docking scores and binding models. The results are shown in Figure 3. Overall, the QuickVina-W scores of the generated molecules outperformed both the other methods and the molecules in the transfer learning dataset, suggesting that PETrans can be able to generate molecules with higher affinity for the target protein. It is also noteworthy that the molecules generated successfully by PETrans (QED, SA score, and LogP) showed significantly better drug potential than the other methods, suggesting that the molecules generated by PETrans are likely to be the correct drug candidates.

### 2.4. Experimental Results with the Dictionary of Protein Secondary Structure (DSSP)

The dictionary of protein secondary structure (DSSP) is a powerful tool for understanding the structural features of proteins, and it has been widely used in a variety of applications, including protein folding, protein design, and drug design. The DSSP uses a combination of algorithms and rules to predict the secondary structure of a protein based on its amino acid sequence [44]. It takes into account the hydrogen bonding patterns between the peptide bonds, as well as the geometry of the protein’s backbone.

We tried to use the PDB file of the target protein to add the structural information of the protein with the DSSP. The results of the experiments are shown in Table 2. The results of the experiments indicate that the incorporation of the DSSP does not significantly enhance docking scores. However, it should be noted that the incorporation of the DSSP may result in a decrease in the QED and SA scores of the generated molecules. Therefore, protein sequences contain sufficient information to support this task.

### 2.5. Experimental Results with Control of the Docking Score

As for ligand interactions, we introduced a docking score control in PETrans. The results of the experiments are shown in Table 3. QuickVina-W was utilized as the docking program, with the same docking parameters and docking files being applied consistently. This ensured that the results were comparable. This resulted in an increase in the docking scores of the molecules generated for EGFR and HTR1A, while the docking scores of the molecules generated for S1PR1 decreased. It was observed that the introduction of a docking score control resulted in a decrease in the validity ratio, novelty ratio, and uniqueness ratio of the generated molecules to varying degrees. The ratio of available molecules should be prioritized in molecule generation tasks. Therefore, this approach is not suitable for our model.

### 2.6. The Distribution of the Docking Scores

We compared the physicochemical properties of the generated molecules to those of the known active and decoy compounds. Each molecule was docked three times and the optimal docking score was selected. Regarding the distribution of docking scores, the docking scores of the 1350 generated molecules showed a good improvement compared to the transfer learning dataset. The minimum calculated docking score was −9.8 kcal/mol. The maximum calculated docking score was −5.9 kcal/mol and the average score was −7.93 kcal/mol. The distribution of docking scores of the generated molecules and the transfer learning dataset are shown in Figure 4. The distributions of the docking scores of the generated molecules for S1PR1 and HTR1A are shown in Figure A4 and Figure A5, respectively. As can be seen in Figure 4b, the molecules generated by PETrans have better docking scores compared to the known active compounds in the transfer learning dataset and the known active and decoy compounds for EGFR from the enhanced directory of useful decoys (DUD-E) [45].

### 2.7. Shifting Distributions of Properties during Transfer Learning

We compare the distributions of four molecular properties of the generated molecules with the transfer learning dataset and the DUD-E dataset (Figure 5). See Figure A6 and Figure A7 for the distributions of the SA scores and the QED scores of the generated molecules for other targets. The average QED of the generated molecules is 0.452 ± 0.05, the average LogP is 4.567 ± 4.17 and the average SA is 2.736 ± 0.57.

### 2.8. Molecular Similarities between the Generated Molecules and the Known Active Compounds

To confirm the novelty of the generated molecules, we calculated the molecular similarity between the generated molecules and the known active compounds. The Morgan fingerprint and similarity scores (Tanimoto score) were calculated using RDKit. The similarity scores ranged from 0 to 1, with scores close to 1 indicating a higher similarity. The similarity distribution of generated molecules is shown in Figure 6. Five pairs of molecules with similarity scores and docking scores are shown in Figure 7. This shows that our model can generate novel compounds similar to the active compounds but with a higher docking score.

## 3. Discussion

The number of drug-like compounds in the chemical space is estimated to be 10^23^–10^60^, making it computationally impossible to fully explore the vast chemical space. Efficient extraction of novel lead compounds from such a large chemical space is a real challenge in drug discovery [46]. Previous studies have successfully evaluated high-throughput screening and virtual screening for large chemical libraries with various filters. With the rapid development of machine learning techniques, quantitative structure activity relationships (QSAR) have become an indispensable virtual screening filter to efficiently and reliably evaluate various physicochemical and pharmacological properties. However, the traditional approach tends to search molecules with desired properties from existing chemical libraries [2,47].

The aim of de novo drug design is to produce a drug that can inhibit the target protein and has balanced physicochemical properties. The limited availability of the small datasets for many tasks can often result in overfitting of powerful models, such as neural networks. Transfer learning is a technique that can address this issue by learning general features in a larger dataset that may also be relevant for a second task with a smaller dataset. The goal of transfer learning is to leverage the knowledge gained from the larger dataset to improve the performance of the model on the smaller dataset. Other methods, such as multi-task learning, also have limitations. While multi-task learning may improve performance on one or two tasks, it may also negatively impact the performance on other tasks. In contrast, transfer learning does not need to balance between tasks, and the primary focus is on the target task. Therefore, transfer learning is used to fine-tune the pretrained model to generate molecules with better binding ability to the target protein.

In this work, we developed a method to design novel molecules for target protein using deep learning. The model used protein encoding and transfer learning to generate novel molecules with better docking scores. Protein encoding is used to extract the physicochemical properties and amino acid information of the target protein. GPT was used to extract the contextual features of the molecule. The model was validated using three target proteins (EGFR, S1PR1 and HTR1A). To test the performance of the model, we performed experiments and compared them with SBMolGen (Table 1). The docking results showed that our model can generate novel molecules with higher docking scores for the target proteins.

There are also some limitations with our model, such as the inability to include docking simulations in the generation process. Including docking simulations in the generation process would allow for the generation of molecules based on the binding affinity and conformation of the target protein. In addition, only the docking score, QED score, SA score, and LogP were used in this study. The inhibitor constant (Ki), cannot be calculated at this stage. In future endeavors, we plan to incorporate docking simulations into molecule generation models. We will also introduce the inhibitor constant (Ki) in our future work to evaluate the experimental and theoretical results.

## 4. Materials and Methods

### 4.1. Datasets

#### 4.1.1. The Drug Dataset (Pretrain)

In this work, we used MOSES [48] as a benchmark dataset. MOSES consists of 1.9 million lead-like molecules from the ZINC dataset with a molecular weight of 250–350 Da, a number of rotatable bonds of less than 7, and an XlogP of less than 3.5 [48]. Additionally, the molecules in the MOSES dataset are small enough to allow further ADMET optimization of the generated molecules [49]. The MOSES dataset is primarily designed to represent lead-like molecules and presents a distribution of molecules with desirable drug-like properties. For pretraining, we split the MOSES dataset into the training set and the test set at a ratio of 9:1.

#### 4.1.2. Target Protein and Active Compounds (Transfer Learning)

Before molecular generation, we prepared EGFR, S1PR1 and HTR1A as target proteins with PDB IDs of 2RGP, 7VIH and 7E2X. The structure of three targets was obtained from the Protein Data Bank (PDB) [50]. In addition, the activate compounds dataset corresponding to three targets were extracted from ExCAPE-DB [51] for transfer learning. This dataset comprises over 70 million SAR data points from publicly available databases (PubChem and ChEMBL) including structure, target information and activity annotations [51]. In transfer learning, we used the same ratio to split the known active compound datasets.

### 4.2. Model Architecture

The architecture of PETrans is divided into three modules: a pretraining module, a transfer learning module, and a docking module. The architecture of PETrans is shown in Figure 8. The protein-encoding process is shown in Figure 9. The core idea of PETrans is to learn the probability distribution of atoms and bond types based on already existing active compounds.

The proposed process of de novo molecule generation is described in detail in the following sections. In Section 4.2.1, we first introduce the pretrained model, explaining the modules included in our model. Section 4.2.2 describes the protein preparation and protein encoding of the target protein. Finally, in Section 4.2.3, we describe the basic parameter settings for the experiments.

#### 4.2.1. The Pretrained Model

The workflow of this study is shown in Figure 8. The protein encoding consists of pseudo amino acid composition (pseAAC), the autocorrelation descriptor (AD), and the conjoint triad (CT). We transform the protein sequences into a 512-dimensional vector by protein encoding. Additionally, then forward them to the SMILES. In certain tasks, such as lead optimization, chemists intend to generate molecules that contain a specific scaffold while achieving the desired property values. So, we used RDkit [52] to calculate the molecular properties and to extract Bemis−Murcko scaffolds [53]. The specific scaffolds are selected to generate molecules targeting the specific target. Each token of the scaffold is mapped to a 256-dimensional vector, using the same embedding layer as SMILES.

The GPT architecture uses a multi-layer Transformer decoder, which is a variant of the transformer [54]. Multi-headed self-attention operations are used in this model, followed by position-wise feedforward layers. This produces an output distribution over target tokens as follows:(1)$$h_{0}=UW_{e}+W_{p}$$
(2)$$h_{l}=transformer\left(h_{l-1}\right)\forall i\in \left[1,n\right]$$
(3)$$P\left(u\right)=softmax\left(h_{n}W_{e}^{T}\right)$$
where $\;U=\left(u_{-k},\ldots ,\;u_{-1}\right)$ is the context vector of tokens, n is the number of layers, $W_{e}$ is the token embedding matrix, and $W_{p}$ is the position embedding matrix.

Self-attention involves three sets of vectors, the query vector, the key vector and the value vector. The weights of the value vectors can be queried using query vectors. Firstly, they are sent through a dot product. A SoftMax function is applied to these vector weights in order to scale the dot products by the dimensionality of the vectors. Each value vector is multiplied by its weight and then summed. A weight matrix is used in each decoder block to calculate the query, key, and value vectors for each token. The attention is expressed by the following equation [55]:(4)$$Attention\left(Q,K,V\right)=softmax(\frac{QK^{T}}{\sqrt{d_{k}}})V$$

We chose MolGPT [56] as the basic GPT model. The model is essentially a mini version of the generative pretraining transformer (GPT) model with only approximately 6 M parameters [56].

To keep track of the input sequence, the model assigns a position value to each token. Segment tokens are provided to distinguish between scaffold and SMILES tokens. Segment tokens indicate whether a particular input is a scaffold or a SMILES molecule, which helps the model distinguish between them. All SMILES tokens in a molecule are mapped from an embedding layer to a 256-dimensional vector. Similarly, a separately trained embedding layer is used to map position tokens and segment tokens to vectors. As input to the model, these embeddings are added to a vector as SMILES token embeddings, position token embeddings, and segment token embeddings.

A molecule is generated by giving the model a start token and sequentially predicting the next token. The starting token is determined using weighted random selection from a list of tokens that appear first in the training set. These tokens are weighted according to the frequency of their occurrence at the first position of the SMILES strings. In the next step, the protein encoding and scaffolds as well as a starting token for the sample of molecules are provided to the model.

#### 4.2.2. Protein Preparation and Encoding

Protein preparation. In this study, EGFR, S1PR1 and HTR1A were used as the target proteins [32,34,35]. The crystal structure of the protein was downloaded and used from the RCSB Protein Data Bank (PDB) dataset. We used AutoDockTools [57] to add hydrogens, compute Gasteiger charges and merge the non-polar hydrogens. Then, the PDB file was converted to pdbqt format. The SMILES was converted to pdbqt format using the program Openbabel [58]. Gasteiger partial charge calculation approach was used to calculate the partial charges of the atoms. Then, we created a 3D grid box for docking.

Selection and extraction of features. We want to fully extract features from protein sequences. Therefore, protein coding methods must include local features, global features, and physicochemical features. PseAAC, AD and CT were used to convert protein sequence into vector and then we integrated three feature extraction methods.

Pseudo amino acid composition (PseAAC) [59] can represent both amino acid composition and amino acid order information. In bioinformatics, it is widely used for protein site prediction, protein subcellular localization prediction, protein structure class prediction, and protein subcellular prediction [60,61,62].

The feature vector of PseAAC is:(5)$$X={\left[x_{1},x_{2},\cdots ,x_{19},x_{20},x_{20+1},\cdots ,x_{20+\varepsilon }\right]}^{T}(\varepsilon <S)$$
where S represents the length of the protein sequence, and the calculation equation of the first 20 elements and the latter ε element in the vector are as:(6)$$x_{u}=\left\{\begin{array}{l} \frac{f_{u}}{\sum_{i=1}^{20}f_{i}+\alpha \sum_{j=1}^{\varepsilon }\theta_{j}},\left(1\leq \mu \leq 20\right)\\ \frac{\alpha \theta_{u-20}}{\sum_{i=1}^{20}f_{i}+\alpha \sum_{j=1}^{\varepsilon }\theta_{j}},\left(21\leq \mu \leq 20+\varepsilon \right) \end{array}\right.$$
where $f_{i}$ is the frequency of occurrence of the number of amino acids in the normalized protein, $\theta_{j}$ is the correlation coefficient of the sequences in the *j*-th layer, and α is the weighting coefficient.

The autocorrelation descriptor (AD) [63] can obtain the physicochemical information contained in the protein sequence. Shen et al. [64] proposed the conjoint triad (CT) method to extract information from protein sequences based on the physicochemical properties of dipoles and volumes [60]. First, 20 amino acids are classified into seven classes based on dipoles and side chain volumes. Considering the interaction between an amino acid and its neighboring amino acids, three contiguous amino acids are considered as one unit, resulting in 7 × 7 × 7 = 343 triad types.
(7)$$d_{i}=\frac{f_{i}-min\left\{f_{1},f_{2},\cdots ,f_{343}\right\}}{max\left\{f_{1},f_{2},\cdots ,f_{343}\right\}},i=1,2,\cdots ,343$$
where $f_{i}$ is the frequency of occurrence of each triad. $d_{i}$ is the 343-dimensional feature vector.

#### 4.2.3. Experimental Setup

The pertained model consists of stacked decoder blocks, including a masked self-attention layer and a fully connected neural network. Each self-attention layer returns a vector of size 256 as input to the fully connected network. The hidden layer of the neural network returns a vector of size 1024, which is activated by the GELU activation layer. The final layer of the fully connected neural network returns a vector of size 256 as input to the next decoder block.

Each model was trained using the Adam optimizer for 10 epochs with a learning rate of 6 × 10^−4^. During generation, the network was provided with a starting token (randomly selected from a list of the first numerator tokens in the training set) and scaffold. An NVIDIA 3090 GPU was used to train and test the models. Most of the models converged after 10 epochs and showed the best performance.

Openbabel was used for generating the 3D structure of the compounds. The hydrogen was added with the -h option and the 3D structure of the molecule was created with the --gen3d option. In the AutoDock Vina program, the grid parameters were set as 44 × 49 × 57 Å^3^ (each for x, y, z dimension, respectively), and 19.496, 35.001, 89.270 (each for x, y, z centers) for the EGFR. For target S1PR1, their corresponding gird parameters were 75 × 75 × 75 Å^3^ and 120.713, 118.886, 131.755. The same is true for target HTR1A. The grid parameters were 75 × 75 × 75 Å^3^ and 93.496, 92.635, 76.821.

## 5. Conclusions

In this study, a new method was developed to design novel molecules for the target protein using deep learning. The model utilized protein encoding and transfer learning to generate novel molecules with high docking scores. The model was validated against three targets (EGFR, S1PR1 and HTR1A). Transfer learning was used to improve the docking scores of the generated molecules. Analysis of the docking scores revealed that the molecules were selective against the active site of the targets. In general, the generated molecules showed better docking scores and broader chemical space distribution than the known active compounds based on the results of an evaluation with three target proteins. In the future, we will introduce docking simulations and the inhibitor constant (Ki) in our work.

## Figures and Tables

**Figure 1 ijms-24-01146-f001:**
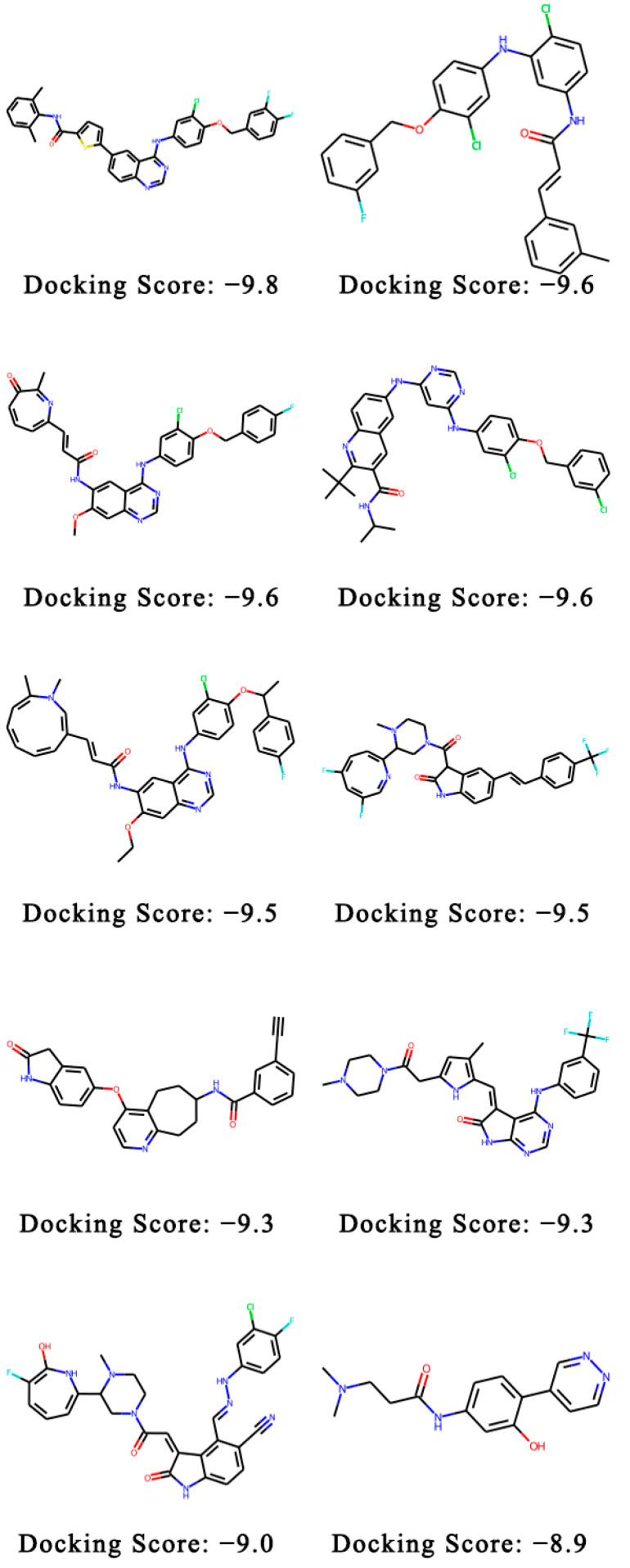
Examples of molecules generated with PETrans against EGFR. Of the molecules generated, those with the best QuickVina-W scores were shown. The colored sections represent different groups.

**Figure 2 ijms-24-01146-f002:**
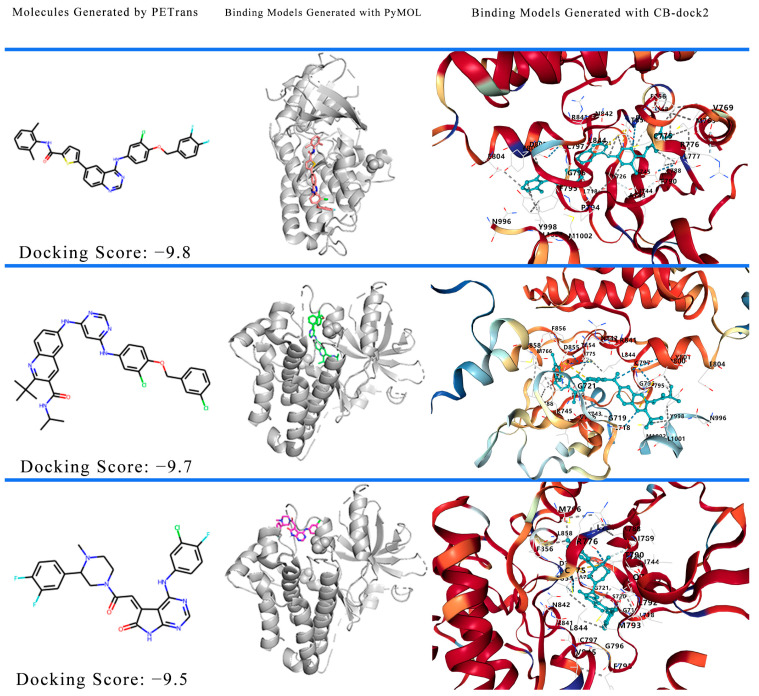
Binding model of the generated molecule. Positions of the generated molecules at the binding sites of the target protein. Each molecule shows the molecular structure (left) and the docked complex structure generated with PyMOL (middle) and CB-dock2 (right), respectively.

**Figure 3 ijms-24-01146-f003:**
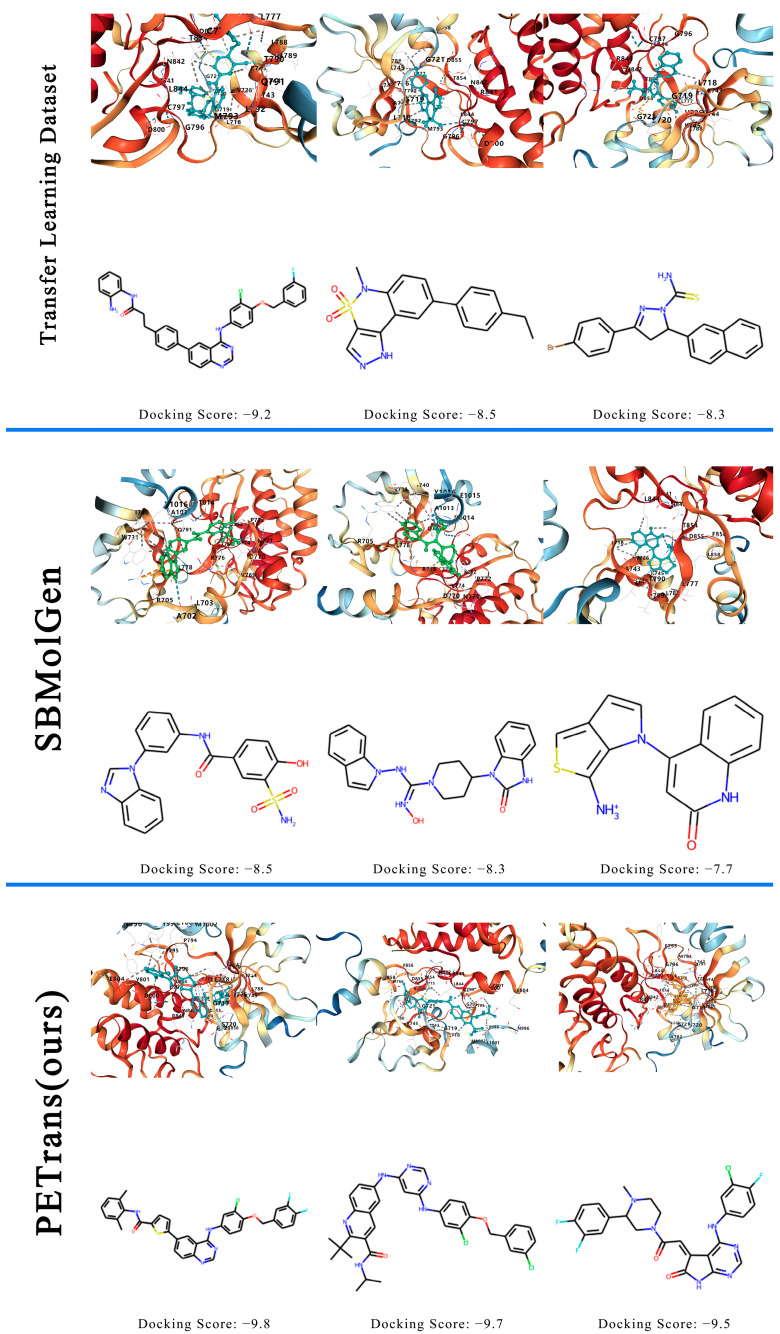
Three examples of molecules in different sets. The comparison of docking scores and binding models of molecules in the transfer learning dataset, SBMolGen and PETrans.

**Figure 4 ijms-24-01146-f004:**
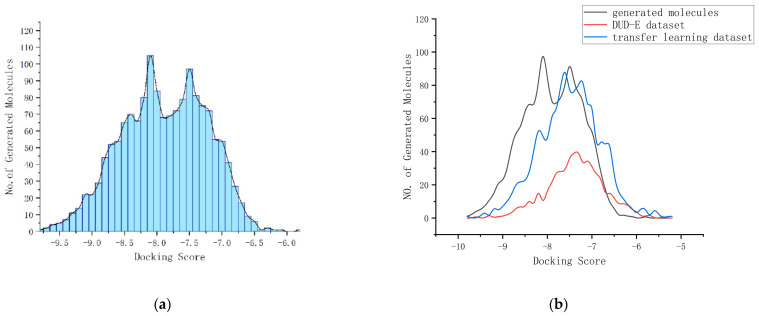
The distribution of docking scores. (**a**) The distribution of docking scores of the generated molecules; (**b**) the distribution of docking scores of the generated molecules and the transfer learning dataset.

**Figure 5 ijms-24-01146-f005:**
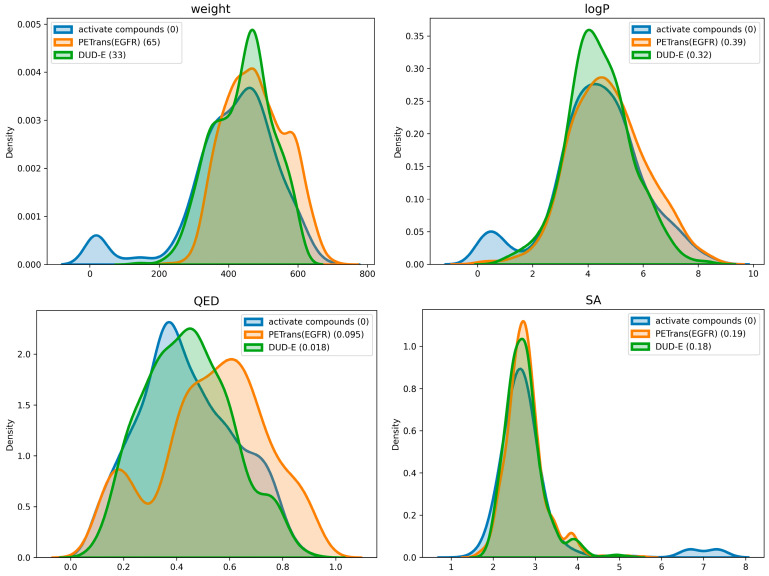
Chemical property distributions for the transfer learning dataset and the generated set of molecules. Wasserstein-1 distances from the transfer learning dataset are given in parentheses. Parameters: MW, LogP, QED, and SA.

**Figure 6 ijms-24-01146-f006:**
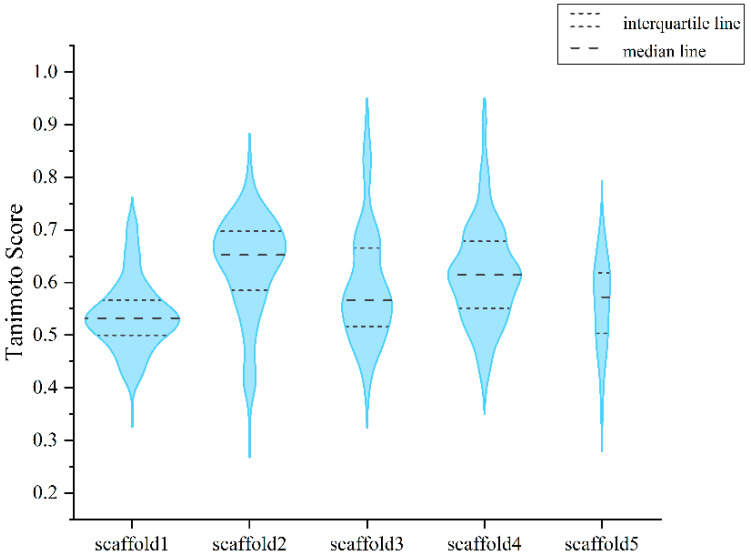
Similarity distribution of generated molecules and known active compounds. Distribution of closest similarity values between the generated compound and the known active compound.

**Figure 7 ijms-24-01146-f007:**
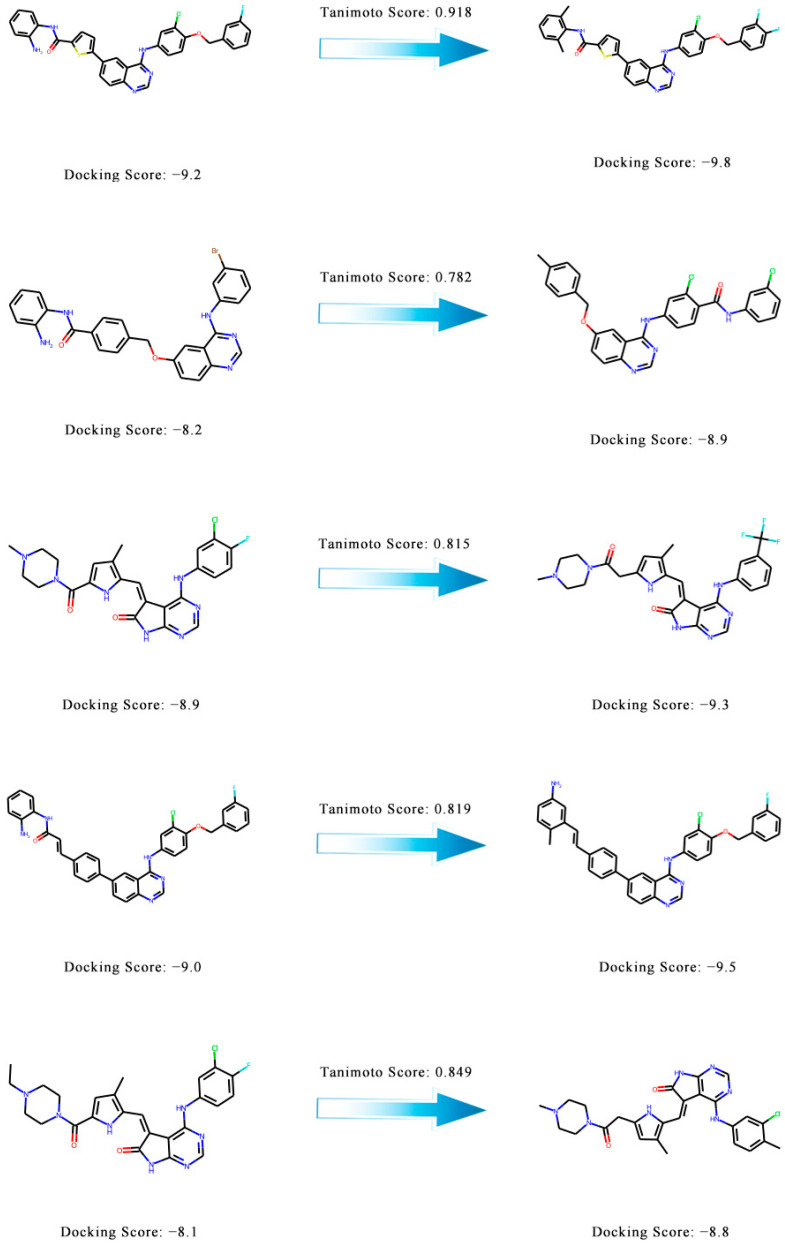
Five pairs of molecules with similarity scores and docking scores. The Tanimoto scores of the known active compounds (left) and the generated molecules (right) are shown above the arrows. The docking scores are shown below the molecular structure.

**Figure 8 ijms-24-01146-f008:**
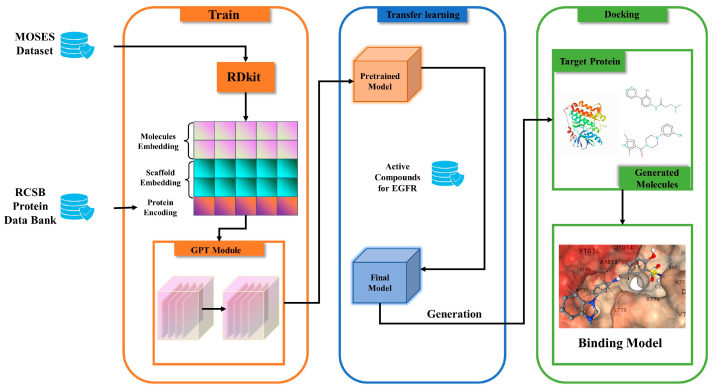
The architecture of PETrans, which consists of three main modules, the pretraining module, the transfer learning module, and the drug-target docking module.

**Figure 9 ijms-24-01146-f009:**
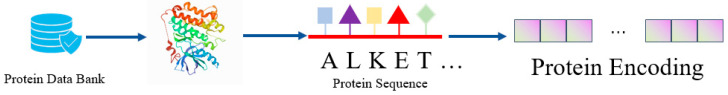
The process of protein encoding. Conversion of protein sequences of different lengths into fixed-length vectors.

**Table 1 ijms-24-01146-t001:** The comparison of the properties of the molecules in the transfer learning dataset and the molecules generated by different methods.

	Docking Score	QED	SA Score	LogP
SBMolGen	−7.416 ± 0.36	**0.509 ± 0.01**	3.233 ± 0.11	4.582 ± 0.88
Transfer Learning Dataset	−7.493 ± 0.46	0.415 ± 0.03	2.769 ± 0.22	4.749 ± 1.90
DUD-E	−7.366 ± 0.42	0.443 ± 0.03	2.747 ± 0.21	4.608 ± 1.35
PETrans (without transfer learning)	−7.689 ± 0.60	0.421 ± 0.03	2.920 ± 0.48	5.185 ± 2.82
PETrans (Ours)	**−7.969 ± 0.58**	0.452 ± 0.05	**2.736 ± 0.57**	**4.567 ± 4.17**

**Table 2 ijms-24-01146-t002:** Experimental results with and without the DSSP.

	Docking Score	QED	SA Score
PETrans (EGFR)	−7.969 ± 0.58	0.452 ± 0.05	2.736 ± 0.57
PETrans (EGFR) *	−8.013 ± 0.60	0.422 ± 0.03	2.921 ± 0.49
PETrans (HTR1A)	−8.589 ± 0.63	0.529 ± 0.02	2.971 ± 0.25
PETrans (HTR1A) *	−8.599 ± 0.61	0.419 ± 0.01	2.267 ± 0.14
PETrans (S1PR1)	−9.579 ± 0.64	0.459 ± 0.02	2.559 ± 0.13
PETrans (S1PR1) *	−9.602 ± 0.60	0.458 ± 0.02	2.561 ± 0.11

* stands for the result with the DSSP.

**Table 3 ijms-24-01146-t003:** Experimental results with and without control of the docking score.

	Docking Score	Valid Ratio	Novelty Ratio	Unique Ratio
PETrans (EGFR)	−7.969 ± 0.58	0.998	1.0	0.953
PETrans (EGFR) *	−8.153 ± 0.52	0.895	1.0	0.719
PETrans (HTR1A)	−8.589 ± 0.63	0.982	1.0	0.979
PETrans (HTR1A) *	−8.784 ± 0.57	0.905	1.0	0.624
PETrans (S1PR1)	−9.579 ± 0.64	0.959	1.0	0.880
PETrans (S1PR1) *	−9.403 ± 0.52	0.815	1.0	0.420

* stands for the result with control of the docking score.

## Data Availability

The datasets mentioned in this article are public datasets. MOSES can be downloaded at: https://github.com/molecularsets/moses (accessed on 18 October 2022). The PDB file and the active compounds can be downloaded at: https://github.com/Chinafor/PETrans (accessed on 18 October 2022).

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
