# Peer review of "PETrans: De Novo Drug Design with Protein-Specific Encoding Based on Transfer Learning"

_ijms, 2023, doi:10.3390/ijms24021146_

Round 1

Reviewer 1 Report

Some activity cliff analysis for near analogs or chemical space map might be of added interest for comparison to these identified top hits to see how far off the activity can be modulated with R-group modification, and how the ML accounts for these possibilities. 

I think it would be useful to mention some pioneering papers where drugs were modeled and refined using ML based techniques and/or advanced simulations (references -- PMID: 34071060, 32160847, 33027969, 31752208, 29416592)

Overall really great manuscript. Few minor grammar things noted. Please re-read and clean up with grammar editor on word program.

Author Response

Firstly, we would like to sincerely thank the editor and the reviewers for their professional work and precious time on our manuscript and for the constructive comments. According to the valuable suggestions, the revised manuscript has been substantially improved. Here are our revisions and reply:

Point 1: Some activity cliff analysis for near analogs or chemical space map might be of added interest for comparison to these identified top hits to see how far off the activity can be modulated with R-group modification, and how the ML accounts for these possibilities. 

Response 1: Thank you for the kind advice. Activity cliffs have originally been defined as pairs of structurally similar compounds that are active against the same target but have a large difference in potency. According to this reference, activity cliffs may be caused by R-group replacements, small chemical changes in core structures, or chiral centers. The potency mentioned here mainly refers to the Compound potency (pKi) values. This value was calculated from the results of wet experiments and requires chemical synthesis and then calculation of the inhibitor constant value. The molecules we generate are not available in existing drug datasets. They have to be chemically synthesized and then calculate the inhibitor constant values. So, we cannot give the Compound potency (pKi) values at this stage.

         As with our model, during the generation process, the network was provided with a starting token (randomly selected from a list of the first numerator tokens in the training set) and a scaffold. The model then sequentially predicts the next token, thus generating a molecule.

From your comments, we learned that when using activity cliff information, connectivity pathways that originate from activity cliffs are more likely to yield highly effective compounds than optimization pathways that originate from other active compounds. We will try it in our future work.

Reference:

Hu, Y.; Stumpfe, D.; Bajorath, J. Advancing the Activity Cliff Concept. F1000Res 2013, 2, 199, doi:10.12688/f1000research.2-199.v1.

Point 2: I think it would be useful to mention some pioneering papers where drugs were modeled and refined using ML based techniques and/or advanced simulations (references -- PMID: 34071060, 32160847, 33027969, 31752208, 29416592)

Response 2: Thank you for providing the references, which were very helpful in revising our article. We have expanded introduction section (Section 1), please kindly refer to line 42-47. We also cited relevant references (references 56-60).

Point 3: Overall really great manuscript. Few minor grammar things noted. Please re-read and clean up with grammar editor on word program.

Response 3: Thanks for the comments. The grammatical errors, references, and equations in the manuscript have been corrected with great caution. We have improved the presentation and English writing.

Reviewer 2 Report

Conventional virtual screening methods sometime inefficient to the discovery of small molecules. The proposed PETrans transfer learning based deep learning method for generating target-specific ligands was proposed. This method is useful when 18
the dataset for target-specific ligands is limited. Generative pre-training (GPT) was used to extract the contextual features of the molecule. Three different protein encoding methods were used to extract the physicochemical properties and amino acid information of the target protein. Protein encoding and molecular sequence information are combined to guide molecule generation. Transfer learning was used to fine-tune the pretrain model to generate molecules with better binding ability
to the target protein. The model was validated using the known cancer target protein, epidermal growth factor receptor (EGFR). The docking results show that the model is capable of generating new molecules with higher docking scores for the target protein.

Specific comments;

1. Validation requires to carry out with DUD data sets.

2. Try structural features and ligand interactions whether it improves the model further

3. abstract requires significant improvement and need appropriate justification of using deep learning method.

4. Validation should include with more literature data sets say with thousands of PDB complexes

5. overall convey of the project should be clear, some rewriting needed

Author Response

Firstly, we would like to sincerely thank the editor and the reviewers for their professional work and precious time on our manuscript entitled “PETrans: De Novo Drug Design with Protein-specific Encoding Based on Transfer Learning” (Manuscript ID: ijms-2040938) and for the constructive comments and supportive recommendations. Here are our revisions and reply:

Point 1: Validation requires to carry out with DUD data sets.

Response 1: Thank you for giving the beneficial suggestion. We have revised Table 1, Figure 4(b) and Figure 5. We added the distribution of the known active and decoy compounds for EGFR from the enhanced directory of useful decoys (DUD-E) dataset. Please refer to line 210-222.

Point 2: Try structural features and ligand interactions whether it improves the model further

Response 2: We propose a general model that can extract the information contained in protein sequences to generate molecules for a specific target protein. However, the limited quantity of known protein structures restricts the application of structure-based prediction methods. Some protein structures are unknown, structural features rely on the structural information predicted by third-party computational tools, which easily result in low efficiency and bring some noise information that impacts the predictive performance. Therefore, we chose the protein sequence to extract protein features.

SBMoleGen used structural information by incorporating docking simulations into the generation process and used the docking score as a reward function for molecular generation. According to the results of the comparison experiment, the molecules generated by PETrans are better than SBMoleGen in terms of docking scores and general properties. The docking scores of the three best molecules of PETrans are -9.8 kcal/mol, -9.7 kcal/mol and -9.5 kcal/mol. Meanwhile, the docking scores of the three best molecules of SBMolGen are -8.5 kcal/mol, -8.3 kcal/mol and -7.7 kcal/mol.

Reference:

Wang, R.; Jin, J.; Zou, Q.; Nakai, K.; Wei, L. Predicting Protein–Peptide Binding Residues via Interpretable Deep Learning. Bioinformatics 2022, 38, 3351–3360, doi:10.1093/bioinformatics/btac352.

Song, B.; Luo, X.; Luo, X.; Liu, Y.; Niu, Z.; Zeng, X. Learning Spatial Structures of Proteins Improves Protein–Protein Interaction Prediction. Brief in Bioinform 2022, 23, doi:10.1093/bib/bbab558.

Point 3: abstract requires significant improvement and need appropriate justification of using deep learning method.

Response 3: Thank you for the comments. We have revised the abstract section (Please refer to line 14-26). We present the main advantages of deep learning methods. A more detailed explanation is given in the introduction section (Please refer to line 73-77).

Here are the details:

Deep learning methods can extract and learn features (representations) in a data-driven way with little or no human participation.

It is possible to generate input vectors from molecular structures, but reversing these vectors is extremely difficult, particularly because a single fingerprint represents multiple possible chemical structures. Using deep generative models with SMILES strings as molecular representations can help overcome this limitation.

Point 4: Validation should include with more literature data sets say with thousands of PDB complexes

Response 4: Thanks for the comments. In the appendix A, we add two validation experiments for PETrans. We applied the method on three targets (EGFR, S1PR1 and HTR1A) to investigate the capability of the PETrans to generate molecules with better binding ability to the target proteins. Related experimental results are shown in FigureA1-A7. Please refer to line 435-456. These results indicate the potential of PETrans for use in future drug design.

In addition, PETrans can also be used to generate molecules for other targets. The protein sequences must be converted into vectors with protein encoding and input to the model together with the SMILES embedding. Transfer learning also requires a dataset of the known active compounds for the specific protein. Then retrain the model and generate molecules for other specific targets.

Point 5: overall convey of the project should be clear, some rewriting needed

Response 5: Thanks for the comments. We have revised the article, removed repetitive statements and rewrote some sentences.

Round 2

Reviewer 2 Report

Point 2. Try structural features and ligand interactions whether it improves the model further

Reviewer wanted to see how it performs for this question.

You may want to use the PDB structural complex for one or few protein target classes and compare your PETrans denovo method to see how it performs.

Also why you have used Transfer learning?? What about other learning? multitasking? did you compared those?

Author Response

Firstly, we would like to sincerely thank the editor and the reviewers for their professional work and precious time on our manuscript entitled “PETrans: De Novo Drug Design with Protein-specific Encoding Based on Transfer Learning” (Manuscript ID: ijms-2040938) and for the constructive comments and supportive recommendations. The WORD file contains details about the experimental results and responses. Please see the attachment.

Round 3

Reviewer 2 Report

Include the whole response within the manuscript in the appropriate sections and also introducing the limitations section.

Author Response

Thank you for the comments.

The experimental results for structural features and ligand interactions have been included in the Results section (Section 2) of the manuscript. Specifically, these results can be found in Section 2.4 and Section 2.5, beginning at line 201.

The revised Discussion section (Section 3) of the manuscript includes a discussion of transfer learning and multi-task learning (Please refer to line 287-297). The limitations of the model are also discussed in the revised Discussion section (Please refer to line 307-314).